# The Effects of Phytase and Non-Starch Polysaccharide-Hydrolyzing Enzymes on Trace Element Deposition, Intestinal Morphology, and Cecal Microbiota of Growing–Finishing Pigs

**DOI:** 10.3390/ani13040549

**Published:** 2023-02-04

**Authors:** Fenfen Liu, Jing Li, Hengjia Ni, Md. Abul Kalam Azad, Kaibin Mo, Yulong Yin

**Affiliations:** 1Key Laboratory of Agro-Ecological Processes in Subtropical Region, Hunan Provincial Key Laboratory of Animal Nutritional Physiology and Metabolic Process, Hunan Research Center of Livestock and Poultry Sciences, South Central Experimental Station of Animal Nutrition and Feed Science in the Ministry of Agriculture, Institute of Subtropical Agriculture, Chinese Academy of Sciences, Changsha 410125, China; 2Department of Animal Science, Hunan Agriculture University, Changsha 410125, China; 3College of Veterinary Medicine, South China Agricultural University, Guangzhou 510642, China

**Keywords:** phytase, non-starch polysaccharide-hydrolyzing enzymes, trace element, gut health, pig

## Abstract

**Simple Summary:**

Phytase and NSPase have been widely used to improve growth performance in swine by improving nutrient utilization. However, the effects of phytase and NSPase (β-glucanase, xylanase, and β-mannanase) with corn–soybean meal-based diets on the trace element deposition of pigs still remains unknown. In this study, the effects of phytase, β-glucanase, xylanase, and β-mannanase on the trace element deposition and intestinal health of growing–finishing pigs were compared. In conclusion, phytase and xylanase supplementation increased the zinc deposition in pigs. Additionally, the supplementation of NSPases may improve the gut health of pigs by modulating the intestinal morphology and microbiota.

**Abstract:**

This study investigated the effects of supplementing phytase and non-starch polysaccharide-degrading enzymes (NSPases) to corn–soybean meal-based diet on the growth performance, trace element deposition, and intestinal health of growing–finishing pigs. Fifty pigs were randomly assigned into the control (basal diet), phytase (basal diet + 100 g/t of phytase), β-mannanase (basal diet + 40 g/t of β-mannanase), β-glucanase (basal diet + 100 g/t of β-glucanase), and xylanase (basal diet + 100 g/t of xylanase) groups. The results show that the supplementation of phytase and NSPases had no impacts (*p*  >  0.05) on the growth performance of pigs. Compared with the control group, pigs fed with xylanase had higher (*p* < 0.05) Zn concentrations in the ileum and muscle and those fed with phytase had higher (*p* < 0.05) Zn concentrations in the ileum. Phytase and xylanase supplementation decreased (*p* < 0.05) fecal Zn concentrations in pigs compared with the control group (*p* < 0.05). In addition, phytase, β-mannanase, β-glucanase, and xylanase supplementation up-regulated (*p* < 0.05) the *FPN1* expression, whereas xylanase up-regulated (*p* < 0.05) the *Znt1* expression in the duodenum of pigs compared with the control group. Moreover, phytase, β-glucanase, and xylanase supplementation up-regulated (*p* < 0.05) the jejunal *Znt1* expression compared with the control group. The intestinal morphology results show that the phytase, β-mannanase, and xylanase groups had increased villus heights (VHs), an increased villus height–crypt depth ratio (VH:CD), and decreased crypt depths (CDs) in the duodenum, whereas phytase, β-mannanase, β-glucanase, and xylanase groups had decreased VH and VH:CD, and increased CD in the jejunum compared with the control group (*p* < 0.05). Pigs fed with exogenous enzymes had decreased bacterial diversity in the cecum. The dietary supplementation of NSPases increased the relative abundance of Firmicutes and decreased spirochaetes (*p* < 0.05). Compared with the control group, dietary NSPase treatment decreased (*p* < 0.05) the opportunistic pathogens, such as *Treponema_2* and *Eubacterium_ruminantium*. Moreover, the relative abundances of *Lachnospiraceae_XPB1014* and Lachnospiraceae were enriched in the β-glucanase and β-mannanase groups (*p* < 0.05), respectively. In conclusion, phytase and xylanase supplementation may promote zinc deposition in pigs. Additionally, the supplementation of NSPases may improve the gut health of pigs by modulating the intestinal morphology and microbiota.

## 1. Introduction

Trace elements such as iron (Fe), copper (Cu), and zinc (Zn) are essential nutrients for the normal growth of animals. Generally, trace elements from the diet are absorbed by the gastrointestinal tract (GIT) and enter into the blood circulation. However, the interactions between trace elements and other substances in diets impaired the absorption of trace elements in the GIT [1]. Therefore, enhancing trace element utilization in diets could be a potential strategy to improve the nutritional quality of diets. Corn and soybean meals are commonly used as feed ingredients for pigs because of their high nutritional values; however, the available trace elements in corn and soybean are limited due to the presence of phytate and non-starch polysaccharides (NSPs) [2,3,4]. Phytate, a hexaphosphoric ester of cyclohexane, consists of insoluble complexes with divalent cations that cannot be absorbed by the small intestine [5]. NSPs in corn and soybean meal are predominantly composed of β-glucans, arabinoxylans, and mannans. The functional properties of some NSPs, such as solubility, viscosity, and water- and ion-binding capacity, can impair mineral absorption in the GIT of pigs [6]. Monogastric animals cannot digest NSPs and phytate due to the lack of endogenous NSP-degrading enzymes (NSPases) and phytases [7]. Therefore, the anti-nutritional effects of NSPs and phytate may improve due to the impact of the supplementation of NSPases and phytase [8,9]. Phytase can break down the phytate molecules and release phosphorus and minerals [10]. NSPases can partially hydrolyze NSPs and thus reduce digesta viscosity and rupture plant cell walls to release cellular nutrients for digestion [4,11]. Thus, NSPases have been mostly applied as feed supplements to promote the growth performance of animals [12,13]. Moreover, the dietary supplementation of NSPases in corn–soybean meal-based diets may produce numerous short-chain oligosaccharides, which may act as prebiotics to modulate intestinal microbiota [14]. Pigs in growing–finishing stage have very large populations, consume large amounts of feed, and excrete the maximum amount of trace elements as slurry [15,16]. However, it is unclear if supplementing β-mannanase, β-glucanase, and xylanase alone in corn–soybean meal-based diets will affect the trace element deposition of growing–finishing pigs. Therefore, the present study was conducted to evaluate the effects of dietary NSPases (β-mannanase, β-glucanase, and xylanase) and phytase supplementation with corn–soybean meal-based diets on the growth performance, trace element deposition, intestinal morphology, and cecal microbiota of growing–finishing pigs.

## 2. Materials and Methods

### 2.1. Animals, Diets, and Experimental Design

The experimental protocol used in the present study was approved by the Animal Care and Use Committee of the Institute of Subtropical Agriculture, Chinese Academy of Sciences (IACUC # 201302).

Fifty healthy crossbred (Landrace × Large White) pigs with an average body weight of 55.95 ± 6.46 kg were selected and randomly assigned to one of five groups with ten replicates (pens) per group and one pig per pen. The five treatment groups included the control (fed a basal diet), phytase (fed a basal diet supplemented with 100 g/t of phytase), β-mannanase (fed a basal diet supplemented with 40 g/t of β-mannanase), β-glucanase (fed a basal diet supplemented with 100 g/t of β-glucanase), and xylanase (fed a basal diet supplemented with 100 g/t of xylanase). Phytase, β-mannanase, β-glucanase, and xylanase were mixed separately with the powder-form basal diet. All pigs had free access to feed and water at all times. The diets were formulated according to the NRC recommendation (NRC, 2012) to meet the nutrient requirements of growing–finishing pigs [17]. The ingredients and calculated nutrient levels of the basal diet are presented in Table 1. The exogenous enzymes (phytase, β-mannanase, β-glucanase, and xylanase) were provided by the Wuhan Sunhy Biology Co., Lid. (Wuhan, China). The phytase, β-mannanase, β-glucanase, and xylanase contained 5000U of phytase/g, 1000 U of β-mannanase/g, 5000 U of β-glucanase/g, and 10,000 U of xylanase/g, respectively.

The animal trial lasted 30 days. Fecal collection was completed in the 30 d. At the end of the experiment, six pigs with an average body weight per group were randomly selected and euthanized under commercial conditions by electrical stunning (250 V, 0.5 A, for 5~6 s). The intestinal (duodenum, jejunum, and ileum), kidney, muscle, and liver tissue samples were collected, immediately frozen into liquid nitrogen, and stored at −80 °C until further analyses. Additionally, cecal digesta were collected and immediately stored at −80 °C for microbiota analysis.

### 2.2. Growth Performance

The pigs were individually weighed at the beginning and end of the experiment. Feed consumption per pen was recorded daily. The average daily gain (ADG), average daily feed intake (ADFI), and feed-to-gain ratio (F/G ratio) were calculated as previously described [18].

### 2.3. Trace Element Analysis

The concentrations of Fe, Cu, and Zn in the jejunum, ileum, muscle, liver, kidney, and fecal were determined by the method described by Nguyen et al. [19]. Briefly, approximately 1.00 g of each sample was accurately weighed into Teflon tubes (Milestone, Sorisole, Bergamo, Italy) and then digested with 8 mL of HNO_3_ and 1 mL of H_2_O_2_ using an UltraWAVE microwave digestion system (Milestone, Sorisole, Bergamo, Italy). The digested samples were diluted to 10 mL with 1% nitric acid to analyze the trace element concentrations using an inductively coupled plasma emission spectrometer (ICP-720ES; Agilent, Palo Alto, CA, USA).

### 2.4. Analysis of mRNA Expression

Quantitative real-time PCR (RT-qPCR) analysis was performed, as described previously [20]. Briefly, the total RNA of the duodenum, jejunum, and ileum was extracted using the TRIZOL reagent (Invitrogen, Carlsbad, CA, USA) and then reversely transcribed into complementary DNA (cDNA) using RevertAid reverse transcriptase (Takara, Otsu, Japan). RT-qPCR analysis was performed with SYBR green I fluorescent dye (Takara, Otsu, Japan). The relative levels of the mRNA expression of the target genes were calculated using the 2^−ΔΔCt^ method [21,22]. All primer sequences are presented in Table 2.

### 2.5. Assessment of Small Intestinal Morphology

The paraformaldehyde-fixed duodenal, jejunal, and ileal samples were dehydrated and embedded in paraffin following the procedures described by Liu et al. [23]. Five µm thick transverse sections were stained with hematoxylin and eosin (H&E). The villus height (VH) and crypt depth (CD) were measured using a light microscope (Leica DMI3000B microscopy, Germany), and the VH-to-CD ratio (VH:CD) was calculated.

### 2.6. Cecal Microbiota Analysis

Cecal microbial DNA was extracted using the stool DNA kit (Omega Bio-Tek, Norcross, GA, USA). The genes of all microbial 16S rRNA sequences in the hypervariable regions of V4-V5 were amplified by PCR using the universal primers 515F 5′-barcode- GTGCCAGCMGCCGCGG)-3′ and 907R 5′-CCGTCAATTCMTTTRAGTTT-3′. The PCR-amplified products were extracted by 1.5% agarose gel electrophoresis, purified using an AxyPrep DNA gel extraction kit (Axygen Biosciences, Union City, CA, USA), and quantified using the QuantiFluor™ ST (Promega, Madison, WI, USA). The purified amplicons were pooled in equimolar from each sample and paired-end and sequenced (2 × 250) on an Illumina MiSeq platform (Illumina, San Diego, CA, USA), according to the standard protocols by Shanghai MajorBio pharm Technology Co., Ltd. (Shanghai, China). The raw reads can be found at: https://dataview.ncbi.nlm.nih.gov/object/PRJNA904409. Raw fastq files were demultiplexed and quality-filtered using QIIME (version 1.17). Operational units (OTUs) were clustered with 97% similarity cutoff using UPARSE (version 7.1) and chimeric sequences were identified and removed using UCHIME. The taxonomy of each 16S rRNA gene sequence was analyzed by the RDP classifier (http://rdp.cme.msu.edu/) against the silva (SSU115)16S rRNA database using a confidence threshold of 70% [24].

### 2.7. Statistical Analysis

Except for microbiome analysis, all statistical analyses were performed using IBM SPSS 22.0 software (SPSS Inc, Chicago, IL, USA). The differences among experimental treatments were performed using one-way ANOVA analysis followed by a Tukey test [25,26]. The Kruskal–Wallis (KW) nonparametric analysis of variance was used to establish the statistical significance of 16S rRNA gene sequencing data. Data are presented as mean value ± standard error of the mean (SEM). Differences were considered significant when *p* < 0.05 and noted with different superscript letters.

## 3. Results

### 3.1. Growth Performance

The growth performance results of growing–finishing pigs are presented in Table 3. Dietary phytase, β-mannanase, β-glucanase, and xylanase supplementation had no significant effects on the growth performance of growing–finishing pigs (*p* > 0.05).

### 3.2. Trace Element Concentrations in Selected Tissues

The results of trace mineral concentrations (Zn, Fe, and Cu) in the jejunum, ileum, liver, kidney, and muscle are presented in Table 4. Phytase, xylanase, and β-mannanase supplementation increased (*p* < 0.05) ileal Zn concentrations, making them higher than those fed with the control diet. The phytase group increased (*p* < 0.05) the Fe concentrations in the kidney and muscle compared with the control group. The concentrations of Zn in the muscle of pigs from the xylanase group were significantly higher than for the control pigs (*p* < 0.05).

### 3.3. Fecal Trace Element Concentrations

The results of fecal trace element concentrations are shown in Table 5. Phytase and xylanase supplementation significantly decreased (*p* < 0.05) the fecal Zn concentrations compared with the control group. In addition, the fecal Fe concentrations significantly decreased (*p* < 0.05) in the phytase group. Meanwhile, exogenous enzyme supplementation had no effect on fecal Cu concentrations (*p* > 0.05).

### 3.4. Expression of Genes Associated with Trace Element Absorption and Tight-Junction Proteins

The relative mRNA levels of trace element transporters in the small intestine are shown in Figure 1. β-mannanase and β-glucanase diets up-regulated (*p* < 0.05) the divalent metal-ion transporter-1 (*DMT1*) expression, while xylanase supplementation up-regulated (*p* < 0.05) the zinc transporter 1 (*Znt1*) expression in the duodenum of pigs compared with the control group. Moreover, compared with the control group, the supplementation of phytase, β-mannanase, β-glucanase, and xylanase up-regulated (*p* < 0.05) the ferroportin 1 (*FPN1*) expression in the duodenum of pigs (Figure 1A). In the jejunum, the supplementation of phytase, β-glucanase, and xylanase up-regulated (*p* < 0.05) the *Znt1* expression (Figure 1B). In the ileum, compared with the control group, phytase and xylanase supplementation down-regulated (*p* < 0.05) the solute carrier family 39-member 4 (*ZIP4*) expression (Figure 1C).

### 3.5. Small Intestinal Morphology

The effects of NSPases and phytase on the duodenal, jejunal, and ileal morphologies of growing–finishing pigs are shown in Table 6 and Figure 2. Dietary phytase, β-mannanase, and xylanase supplementation increased (*p* < 0.05) VH and VH:CD, and decreased (*p* < 0.05) CD in the duodenum of pigs compared with the control group. In addition, dietary phytase, β-mannanase, β-glucanase, and xylanase supplementation decreased (*p* < 0.05) VH and VH:CD in the jejunum of pigs.

### 3.6. Expression of Genes Associated with Tight-Junction Proteins

As shown in Figure 3, compared with the control group, the supplementation of xylanase up-regulated (*p* < 0.05) the zonula occludens-1 (*ZO-1*) expression in the duodenum (Figure 3A). In addition, compared to the β-mannanase, the xylanase diet up-regulated (*p* < 0.05) the *claudin-1* mRNA expression in the duodenum (Figure 3A). Meanwhile, exogenous enzyme supplementation had no effect on the relative mRNA levels of tight-junction proteins in the jujenum and ileum (*p* > 0.05).

### 3.7. Cecal Microbiota Analysis

A total of 1,128,688 valid reads were generated from the 30 samples through MiSeq sequencing analysis. Dietary phytase, β-glucanase, and xylanase supplementation decreased (*p* < 0.05) the Shannon index in the cecum of pigs compared with the control group (Table 7). Moreover, compared with the control group, the supplementation of phytase, β-glucanase, and xylanase increased (*p* < 0.05) the Simpson index (Table 7).

The six most dominant phyla of the microbial community among the different treatment groups were Firmicutes, Bacteroridetes, Proteobacteria, Verrucomicrobia, Spirochaetes, and Tenericutes (Figure 4A). Compared with the control group, the relative abundance of Firmicutes was higher (*p* < 0.05) in the β-mannanase, β-glucanase, and xylanase groups, whereas the abundance of Spirochaetes was lower (*p* < 0.05) in the phytase, β-mannanase, β-glucanase, and xylanase groups. Compared with the control group, dietary phytase and β-mannanase supplementation also lowered (*p* < 0.05) the relative abundance of Tenericutes (Figure 5A). At the genus level, *Terisporobacter*, *Lactobacillus*, *Eubacterium_copostanoligenes*, and *Clostridium_sensu_stricto_1* were the top four abundant genera (Figure 4B). Compared with the control group, dietary phytase supplementation increased (*p <* 0.05) the relative abundances of *Romboutsia* and decreased (*p* < 0.05) the relative abundances of *Ruminococcaceae_UCG-014*, *Prevotellaceae_NK3B31*, *Eubacterium_ruminantium*, and *Oscillospira* in the cecum of pigs. β-mannanase supplementation significantly increased (*p < 0.05*) the relative abundances of *Romboutsia* and decreased (*p <* 0.05) the relative abundances of *Eubacterium_ruminantium* and *Oscillospira* compared with the control group. Compared with the control group, dietary β-glucanase supplementation significantly increased (*p <* 0.05) the relative abundances of *Romboutsia* and *Lachnospiraceae_XPB1014* and decreased the relative abundances of *Prevotellaceae_NK3B31*, *Eubacterium_ruminantium*, and *Oscillospira*. Moreover, the relative abundances of *Eubacterium_ruminantium* and *Oscillospira* were decreased (*p < 0.05*) in the xylanase group (Figure 5B).

The effects of NSPases and phytase on cecal microbiota in pigs were also identified using linear discriminant analysis (LDA) and effect size (LefSe) analysis (LDA score > 4) (Figure 6). The genera *Lachnospiraceae_XPB1014* and *Romboutsia* were enriched in the β-glucanase group, the family Lachnospiraceae was enriched in the β-mannanase group, and the phylum Firmicutes was enriched in the xylanase group. However, the families *Prevotellaceae_NK3B31*, *Ruminococcaceae_UCG-005*, and *Spirochaetae* were enriched in the control group. Moreover, the phylum Spirochaetes, the class Spirochaetes, the order Spirochaetales, the family Spirochaetaceae, and the genus *Treponema_2* were also enriched in the control group.

## 4. Discussion

In the present study, dietary NSPases and phytase had no impacts on the growth performance of growing–finishing pigs, indicating that the dietary phytase, β-mannanase, β-glucanase, and xylanase used in this study did not have negative effects on the growth rate of the pigs. Although phytase, β-mannanase, β-glucanase, and xylanase supplementation alone did not affect the growth performance parameters of growing–finishing pigs, there is still a question of whether these supplements can influence the trace element deposition or not. To evaluate the deposition of trace elements in pigs, we determined the Fe, Cu, and Zn concentrations in the jejunum, ileum, liver, kidney, and muscle. The results show that dietary xylanase supplementation increased the Zn concentrations in the ileum and muscle of pigs. In addition, the supplementation of phytase increased the Fe content in the kidney, muscle, and Zn content in ileum compared with the control group. Zn, Fe, and Cu excretion as feces were also determined in this study. Dietary phytase supplementation decreased fecal Zn and Fe concentrations. Xylanase supplementation decreased fecal Zn concentrations. We also found that phytase supplementation up-regulated the *Znt1* expression in the jejunum of pigs, while β-mannanase up-regulated *DMT1* expression in the duodenum and β-glucanase up-regulated *DMT1* in the duodenum and *Znt1* expression in the jejunum. In addition, xylanase supplementation up-regulated the *Znt1* expression in the duodenum and jejunum of pigs. Collectively, these results indicate that exogenous enzymes up-regulated the expressions of trace element transporters in the intestine, suggesting that enzymes differentially affected Fe and Zn absorption rates from the small intestine. Thus, the increased trace element concentrations in tissues may be necessary so that more trace elements can be absorbed through the GIT. Phytic acid chelates important cations, forming insoluble complexes in the upper GIT. The use of phytase can reduce the negative effects of phytic acid on the utilization of trace elements [2,27]. Dietary fibers negatively affect the absorption of trace elements in the GIT due to mineral binding or physical entrapment [3]. β-mannanase, β-glucanase, and xylanase can degrade the cell wall structure of polysaccharides in the diet and reduce or eliminate the encapsulation effect of the cell wall polysaccharides, leading to the absorption of more trace elements through the intestine. Our results are consistent with a previous investigation which reported that the treatment of plant-based foods with exogenous enzymes showed beneficial effects on Fe and Zn bioavailability [28,29]. Those results indicate that phytase and xylanase may improve Zn deposition by regulating the gene expressions of trace element transporters and reducing the excretion of fecal trace elements.

Intestinal morphology and gut barriers significantly impact nutrient absorption [30]. In the present study, a histological examination of the small intestine revealed that dietary phytase, β-mannanase, β-glucanase, and xylanase had a positive effect on the duodenal morphology among the pigs in the different treatment groups, i.e., in the form of increased VH and VH:CD and decreased CD in all treatment groups. In addition, we observed that jejunal VH and VH:CD values were lower for the phytase, β-mannanase, β-glucanase, and xylanase groups than the control group. We also found that compared with the control group, phytase supplementation increased the ileal VH. Luo et al. [31] observed that dietary xylanase supplementation showed higher intestinal fold heights and microvillus heights of large yellow croakers. Similarly, Moita et al. [32] found that the dietary supplementation of phytase increased the VH and villus width of the broiler chickens. A previous study by Wu et al. [33] reported that dietary xylanase and phytase reduced the size of the duodenal villi height of broiler chickens fed with a wheat-based diet. The discrepancies in these results might be attributable to fact that the pigs fed corn–soybean meal-based diets supplemented with NSPases increased the production of soluble NSPs in the intestine. Tight junctions constitute intestinal barrier function [34]. Three of the key proteins of the tight junctions are *ZO-1*, *occludin*, and *claudin-1*. The improved expression of tight junctions can enhance the intestinal barrier function [35]. Our results show that the mRNA levels of *ZO-1* were increased in pigs fed xylanase diet. This suggests that xylanase contributed to improving the integrity of the intestinal barrier in growing–finishing pigs.

Another possible reason for increased trace element utilization in the presence of exogenous enzymes is that exogenous enzymes can modulate undigested substrates and alter the gut microbiota composition. The supplementation of phytase and NSPases has been shown to change the composition or number of microbes in animals [36,37]. In the present study, the microbial diversity indices indicate that pigs fed with dietary phytase, β-mannanase, β-glucanase, and xylanase diets had lower Shannon and higher Simpson indices for cecal microbiota than the pigs fed with the basal diet. These results indicate that the use of exogenous enzymes sufficiently changed the conditions of the gastrointestinal environment. At the phylum level, Firmicutes was the dominant phylum in both control and treatment groups, followed by Bacteroidetes, Proteobacteria, Verrucomicrobia, and Spirochaetes. Firmicutes was the major phylum of Gram-positive bacteria, which has been found to be involved in energy resorption and obesity [38,39]. A previous study reported that a diet supplemented with high fibers decreased the abundance of bacteria from the phylum Firmicutes [40]. In the present study, dietary β-mannanase, β-glucanase, and xylanase supplementation significantly increased the abundance of Firmicutes, indicating that the dietary supplementation of NSPases to pigs could effectively hydrolyze fibers and thus limit substrates for cecal microbial fermentation. Another significant change in microbiota abundance was found in phylum Spirochaetes, which are helical, and some of them are pathogenic to humans and animals [41]. Notably, in the present study, we observed a decrease in the abundance of the phylum Spirochaetse, the class Spirochaetes, the order Spirochaetales, and the family Spirochaetaceae in the enzyme-treated groups. At the genus level, opportunistic pathogens, such as *Treponema_2* and *Eubacterium ruminantium*, were also decreased in the NSPases groups compared with the control group. These results indicate that the supplementation of exogenous enzymes could decrease harmful bacteria in the cecum of pigs. *Lachnospiraceae* is also associated with maintaining gut health [42]. In the present study, the genus *Lachnospiraceae_XPB1014* was increased in the β-glucanase group, while the family Lachnospiraceae was increased in the β-mannanase group. This is probably due to the fact that the supplementation of exogenous enzymes could improve the digestion of NSPs in the small intestine and reduce the amount of substrate that is available for putrefactive and starch-utilizing bacteria in the large intestine.

## 5. Conclusions

In summary, the dietary supplementation of phytase and xylanase have positive effects on Zn deposition in pigs, as demonstrated by improving Zn concentrations, regulating the gene expressions of trace element transporters, and decreasing fecal Zn concentrations. In addition, dietary phytase, β-mannanase, β-glucanase, and xylanase altered the intestinal morphology and decreased the microbial richness indices and the relative abundances of opportunistic pathogens. The present study provides theoretical support for applying exogenous enzymes to improve trace element utilization and regulate gut health in livestock production.

## Figures and Tables

**Figure 1 animals-13-00549-f001:**
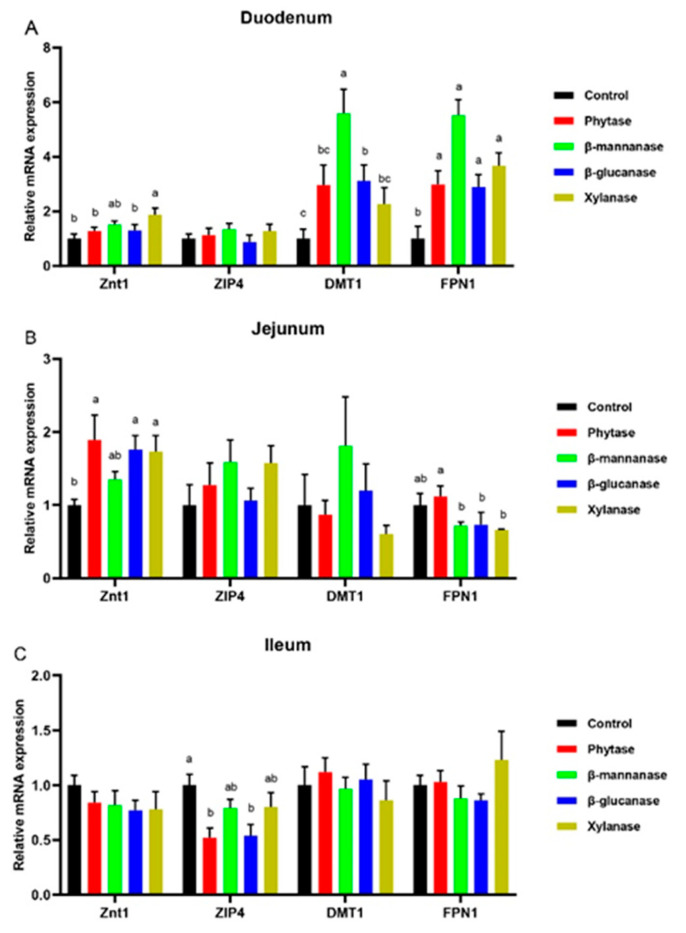
Gene expression levels associated with trace element absorption (*Znt1*, *ZIP4*, *DMT1*, and *FPN1*) in the duodenum (**A**), jejunum (**B**), and ileum (**C**). Control, basal diet; phytase, basal diet + 100 g/t of phytase; β-mannanase, basal diet + 40 g/t of β-mannanase; β-glucanase, basal diet + 100 g/t of β-glucanase; xylanase, basal diet + 100 g/t of xylanase. Data are expressed as mean ± SEM (n = 6). Means within with different superscripts are significantly different (*p* < 0.05). *DMT1*, divalent metal-ion transporter-1; *ZIP4*, solute carrier family 39 member 4; *FPN1*, ferroportin 1; *ZnT1*, zinc transporter 1.

**Figure 2 animals-13-00549-f002:**
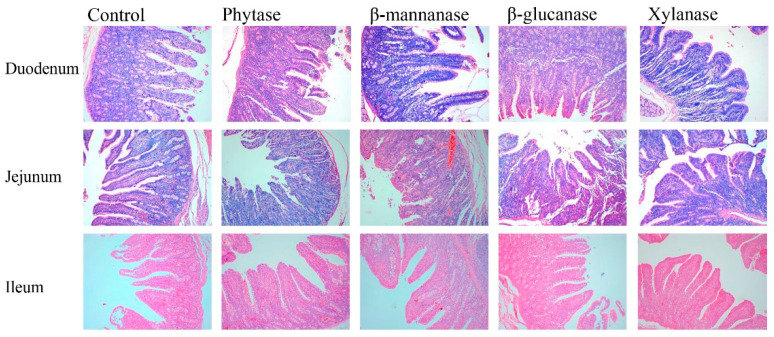
The intestinal morphology was histologically analyzed by hematoxylin and eosin (HE, 500 µm). Control, basal diet; phytase, basal diet + 100 g/t of phytase; β-mannanase, basal diet + 40 g/t of β-mannanase; β-glucanase, basal diet + 100 g/t of β-glucanase; xylanase, basl diet + 100 g/t of xylanase.

**Figure 3 animals-13-00549-f003:**
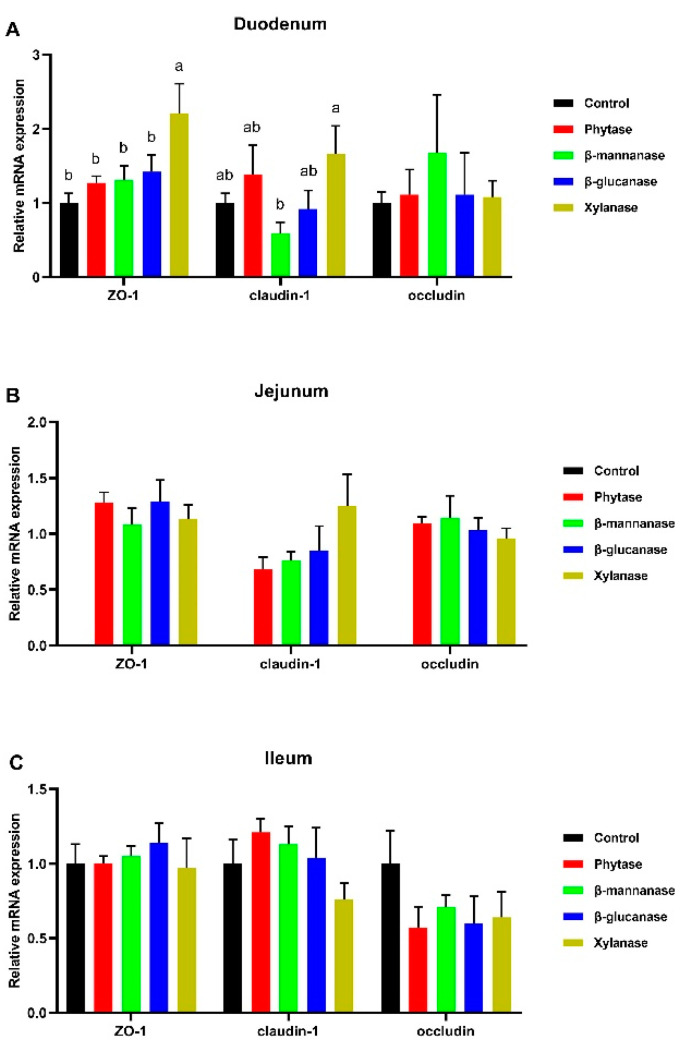
Gene expression levels of the tight-junction proteins (*ZO-1*, *claudin-1*, and *occludin*) in the duodenum (**A**), jejunum (**B**), and ileum (**C**). Control, basal diet; phytase, basal diet + 100 g/t of phytase; β-mannanase, basal diet + 40 g/t of β-mannanase; β-glucanase, basal diet + 100 g/t of β-glucanase; xylanase, basal diet + 100 g/t of xylanase. Data are expressed as mean ± SEM (n = 6). Means within with different superscripts are significantly different (*p* < 0.05). *ZO-1*, zonula occludens-1.

**Figure 4 animals-13-00549-f004:**
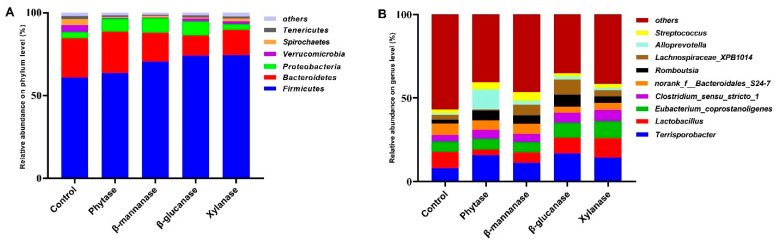
The effects of dietary NSPases and phytase on microbiota composition at the phylum and genus levels of growing–finishing pigs. (**A**) Relative contribution of the top 6 phyla in each group. (**B**) Relative contribution of the top 10 genera in each group. Control, basal diet; phytase, basal diet + 100 g/t of phytase; β-mannanase, basal diet + 40 g/t of β-mannanase; β-glucanase, basal diet + 100 g/t of β-glucanase; xylanase, basal diet + 100 g/t of xylanase.

**Figure 5 animals-13-00549-f005:**
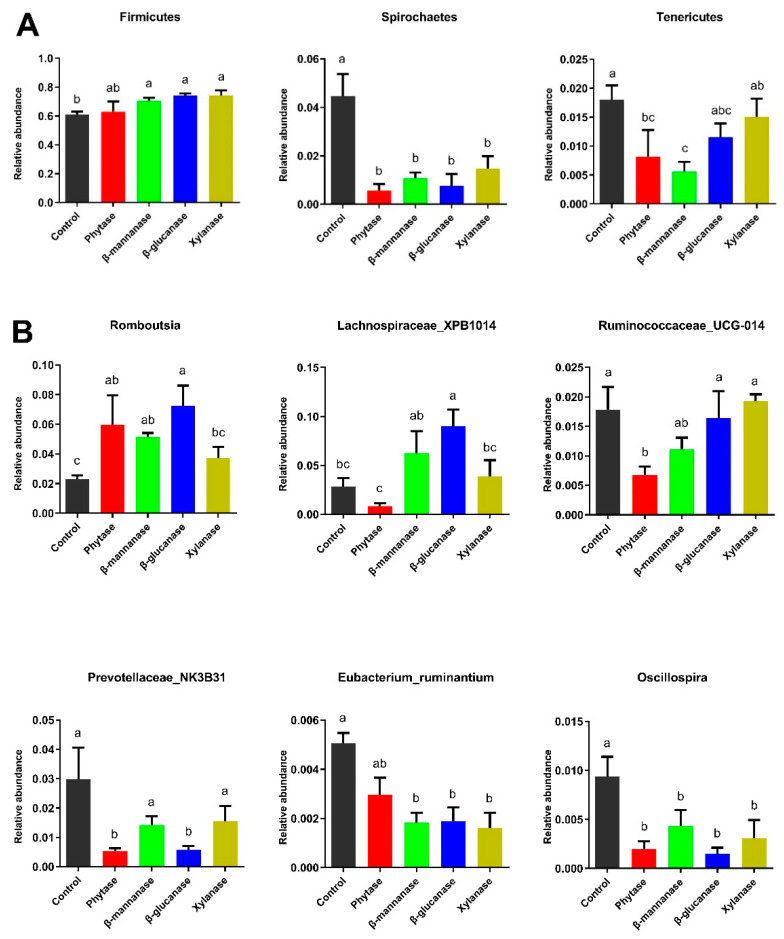
Taxonomic comparison of the cecal microbial community of growing–finishing pigs from the control, phytase, β-mannanase, β-glucanase, and xylanase groups. Differences in the microbial communities at (**A**) the phylum level and (**B**) the genus level. Data are expressed as mean ± SEM (n = 6). Means within a row with different superscripts are significantly different (*p* < 0.05). Control, basal diet; phytase, basal diet + 100 g/t of phytase; β-mannanase, basal diet + 40 g/t of β-mannanase; β-glucanase, basal diet + 100 g/t of β-glucanase; xylanase, basal diet + 100 g/t of xylanase.

**Figure 6 animals-13-00549-f006:**
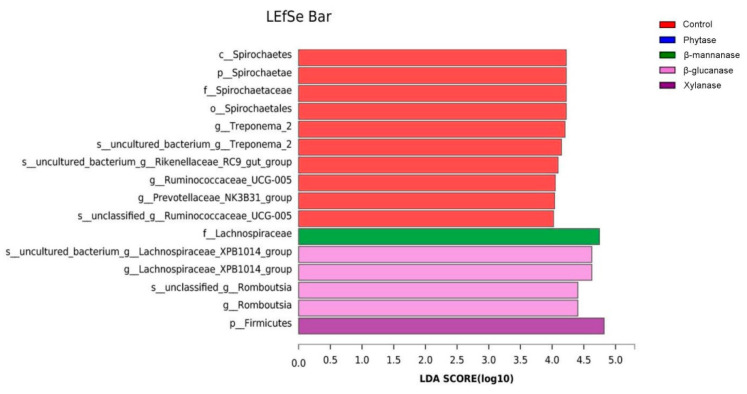
LefSe analysis of the cecal microbial community of growing–finishing pigs from the control, phytase, β-mannanase, β-glucanase, and xylanase groups. Species with significant differences have an LDA score greater than the estimated value with a default score 4. The length of the histogram represents the LDA score, indicating the differences in species among the five groups. Control, basal diet; phytase, basal diet + 100 g/t of phytase; β-mannanase, basal diet + 40 g/t of β-mannanase; β-glucanase, basal diet + 100 g/t of β-glucanase; xylanase, basal diet + 100 g/t of xylanase.

**Table 1 animals-13-00549-t001:** Composition of the basal diet (on an as-fed basis) for pigs.

Items	Inclusion Rate (%)	Calculated Nutrient Levels	
Corn	58.60	Digestible energy (MJ/kg)	14.20
Soybean meal (43% crude protein)	29.00	Crude Protein (%)	18.18
Wheat bran	7.80	Lys (%)	0.98
Soy oil	1.55	Met + Cys (%)	0.55
L-Lys HCl (98%)	0.18	Thr (%)	0.59
Thr	0.10	Trp (%)	0.19
Calcium hydrophosphate	0.69	Fe (mg/kg)	118.0
Limestone	0.87	Cu (mg/kg)	20.0
Salt	0.30	Zn (mg/kg)	80.3
1% Premix ^1^	1.00		
Total	100		

^1^ Premix provided the following per kg of diet: vitamin D_3_ (40 IU), vitamin K_3_ (0.5 mg), vitamin B_1_ (1.0 mg), vitamin A (800 IU), vitamin E (10 IU), vitamin B_2_ (3.6 mg), vitamin B_6_ (1.5 mg), biotin (0.05 mg), folic acid (0.3 mg), D-pantothenic acid (10.0 mg), nicotinic acid (10.0 mg), choline (300.0 mg), Fe as FeSO_4_·H_2_O (30.0 mg), Zn as ZnSO_4_ (50.0 mg), Cu as CuSO_4_·5H_2_O (10.0 mg), I as KI (0.14 mg), Mn as MnSO_4_·H_2_O (80.0 mg), Se as Na_2_SeO_3_ (0.30 mg), Co as CoCl_2_ (1.0 mg).

**Table 2 animals-13-00549-t002:** Primers used in this study.

Genes	Accession No.	5′-3′ Primer Sequences
*DMT1*	NM_001128440.1	R: AGGGAATGTTCCTGCCATCG
		F: ACTGGTGGCTTCTTCAGTCAG
*ZIP4*	XM_021090449	F: TGCTGAACTTGGCATCTGGG
		R: CGCCACGTAGAGAAAGAGGC
*FPN1*	XM_003483701.4	R: TACCAACGGGGTACTTTGCC
		F: AGTGGGGAATGCAATTCAGGA
*ZnT1*	NM_001139470	F: CCAGGGGAGCAGGGAACCGA
		R: TCAGCCCGTTGGAGTTGCTGC
*ZO-1*	XM_021098896.1	F: CCTGCTTCTCCAAAAACTCTT
		R: TTCTATGGAGCTCAACACCC
*occludin*	NM_001163647.2	F: ACGAGCTGGAGGAAGACTGGATC
		R: CCCTTAACTTGCTTCAGTCTATTG
*claudin-1*	NM_001244539.1	F: AAGGACAAAACCGTGTGGGA
		R: CTCTCCCCACATTCGAGATGATT
*β-actin*	XM_003357928	F: CGTTGGCTGGTTGAGAATC
		R: CGGCAAGACAGAAATGACAA

F, forward; R, reverse; *DMT1*, divalent metal-ion transporter-1; *ZIP4*, solute carrier family 39 member 4; *FPN1*, ferroportin 1; *ZnT1*, zinc transporter 1; *ZO-1*, zonula occludens-1.

**Table 3 animals-13-00549-t003:** The effects of dietary NSPases and phytase on the growth performance of growing–finishing pigs.

Items	Control	Phytase	β-Mannanase	β-Glucanase	Xylanase	*p*-Values
ADG (kg)	0.85 ± 0.07	0.94 ± 0.08	0.92 ± 0.13	0.94 ± 0.12	0.92 ± 0.12	0.583
ADFI (kg)	2.45 ± 0.12	2.66 ± 0.38	2.64 ± 0.32	2.71 ± 0.21	2.67 ± 0.35	0.599
F/G ratio	2.88 ± 0.16	2.82 ± 0.28	2.90 ± 0.34	2.92 ± 0.38	2.92 ± 0.24	0.973

Data are expressed as means ± SEM (n = 10). ADG, average daily gain; ADFI, average daily feed intake; F/G, feed-to-gain ratio; control, basal diet; phytase, basal diet + 100 g/t of phytase; β-mannanase, basal diet + 40 g/t of β-mannanase; β-glucanase, basal diet + 100 g/t of β-glucanase; xylanase, basal diet + 100 g/t of xylanase.

**Table 4 animals-13-00549-t004:** The effects of dietary NSPases and phytase on tissue trace element concentrations in growing–finishing pigs (freeze-dry basis, mg/kg).

Items	Control	Phytase	β-Mannanase	β-Glucanase	Xylanase	*p*-Values
Jejunum						
Zn	17.68 ± 1.76	14.65 ± 0.62	15.41 ± 1.91	15.97 ± 1.12	16.47 ± 0.75	0.580
Fe	80.04 ± 11.45	84.37 ± 9.37	59.03 ± 4.68	74.52 ± 8.69	87.86 ± 16.22	0.270
Cu	2.78 ± 0.13	2.37 ± 0.16	2.86 ± 0.36	2.86 ± 0.25	2.99 ± 0.32	0.517
Ileum						
Zn	19.31 ± 0.99 ^c^	22.13 ± 0.93 ^ab^	22.41 ± 0.47 ^ab^	19.88 ± 0.74 ^bc^	23.09 ± 0.99 ^a^	0.045
Fe	54.28 ± 5.37	49.47 ± 5.53	54.96 ± 6.95	49.82 ± 8.33	46.00 ± 2.65	0.825
Cu	1.92 ± 0.06	2.11 ± 0.35	1.76 ± 0.05	1.83 ± 0.20	2.41 ± 0.50	0.428
Liver						
Zn	64.04 ± 10.32	64.62 ± 7.81	61.12 ± 4.99	56.04 ± 4.52	56.99 ± 4.10	0.850
Fe	175.7 ± 26.12	189.7 ± 28.61	119.33 ± 20.27	151.35 ± 11.67	180.7 ± 18.13	0.182
Cu	9.29 ± 0.54	9.76 ± 1.25	9.48 ± 0.74	11.25 ± 1.71	10.74 ± 1.53	0.760
Kidney						
Zn	25.7 ± 3.11	28.9 ± 2.19	24.44 ± 1.13	28.11 ± 1.04	31.18 ± 4.08	0.399
Fe	59.26 ± 2.50 ^b^	74.13 ± 4.01 ^a^	63.79 ± 4.95 ^ab^	72.17 ± 5.18 ^ab^	62.52 ± 5.72 ^ab^	0.030
Cu	10.5 ± 1.12	7.91 ± 0.59	8.37 ± 0.82	9.03 ± 0.72	10.24 ± 1.39	0.278
Muscle						
Zn	11.55 ± 0.67 ^b^	12.35 ± 0.88 ^ab^	13.86 ± 1.12 ^ab^	14.44 ± 1.43 ^ab^	15.55 ± 0.88 ^a^	0.049
Fe	17.90 ± 2.32 ^b^	31.23 ± 3.05 ^a^	24.16 ± 2.15 ^ab^	18.42 ± 2.34 ^b^	22.93 ± 3.30 ^ab^	0.013
Cu	0.62 ± 0.04	0.62 ± 0.08	0.69 ± 0.08	0.67 ± 0.07	0.69 ± 0.06	0.879

Data are expressed as means ± SEM (n = 6). Means within a row with different superscripts are significantly different (*p* < 0.05). Zn, zinc; Fe, iron; Cu, copper; control, basal diet; phytase, basal diet + 100 g/t of phytase; β-mannanase, basal diet + 40 g/t of β-mannanase; β-glucanase, basal diet + 100 g/t of β-glucanase; xylanase, basal diet + 100 g/t of xylanase.

**Table 5 animals-13-00549-t005:** The effects of dietary NSPases and phytase on fecal trace element concentrations in growing–finishing pigs (dry basis, g/kg).

Items	Control	Phytase	β-Mannanase	β-Glucanase	Xylanase	*p*-Values
Zn	1.06 ± 0.08 ^a^	0.97 ± 0.07 ^b^	1.01 ± 0.08 ^ab^	1.00 ± 0.03 ^ab^	0.95 ± 0.03 ^b^	0.037
Fe	1.75 ± 0.34 ^a^	1.23 ± 0.12 ^b^	1.56 ± 0.49 ^a^	1.64 ± 0.51 ^a^	1.84 ± 0.60 ^a^	0.002
Cu	0.97 ± 0.09	0.94 ± 0.03	0.93 ± 0.04	0.92 ± 0.04	0.93 ± 0.10	0.052

Data are expressed as means ± SEM (n = 6). Means within a row with different superscripts are significantly different (*p* < 0.05). Zn, zinc; Fe, iron; Cu, copper; control, basal diet; phytase, basal diet + 100 g/t of phytase; β-mannanase, basal diet + 40 g/t of β-mannanase; β-glucanase, basal diet + 100 g/t of β-glucanase; xylanase, basal diet + 100 g/t of xylanase.

**Table 6 animals-13-00549-t006:** The effects of dietary NSPases and phytase on the duodenal, jejunal, and ileal morphologies of growing–finishing pigs.

Item	Control	Phytase	β-Mannanase	β-Glucanase	Xylanase	*p*-Values
Duodenum						
VH, μm	391.4 ±15.61 ^c^	441.1 ± 23.85 ^a^	435.2 ±23.13 ^ab^	407.8 ± 11.33 ^bc^	423.2 ± 34.52 ^ab^	0.032
CD, μm	249.4 ± 12.4 ^a^	201.8 ± 12.99 ^c^	225.2 ±17.17 ^b^	216.4 ± 20.15 ^bc^	211.1 ± 16.28 ^bc^	0.027
VH:CD	1.56 ± 0.12 ^c^	2.30 ± 0.17 ^a^	1.93 ± 0.13 ^b^	1.88 ± 0.11 ^b^	2.01 ± 0.19 ^b^	0.005
Jejunum						
VH, μm	453.6 ± 30.96 ^a^	414.9 ± 35.54 ^b^	356.4 ± 24.16 ^c^	391.2 ± 35.44 ^bc^	396.8 ± 23.83 ^b^	0.004
CD, μm	194.7 ± 25.10 ^c^	210.0 ± 17.38 ^bc^	215.8 ± 20.05 ^b^	243.6 ± 34.10 ^a^	238.6 ± 21.86 ^ab^	0.039
VH:CD	2.33 ± 0.20 ^a^	1.98 ± 0.21 ^b^	1.65 ± 0.18 ^c^	1.61 ± 0.25 ^c^	1.66 ± 0.19 ^c^	0.021
Ileum						
VH, μm	282.1 ± 30.85 ^c^	334.3 ± 31.32 ^ab^	302.5 ± 48.12 ^bc^	292.0 ± 27.56 ^bc^	352.0 ± 28.17	0.047
CD, μm	229.1 ± 21.85	239.8 ± 19.15	232.0 ± 27.05	228.7 ± 16.82	248.2 ± 25.53	0.771
VH:CD	1.23 ± 0.18	1.38 ± 0.16	1.30 ± 0.18	1.28 ± 0.15	1.42 ± 0.17	0.425

Data are expressed as means ± SEM (n = 6). Means within a row with different superscripts are significantly different (*p* < 0.05). VH, villus height; CD, crypt depth; VH:CD, VH-to-CD ratio; control, basal diet; phytase, basal diet + 100 g/t of phytase; β-mannanase, basal diet + 40 g/t of β-mannanase; β-glucanase, basal diet + 100 g/t of β-glucanase; xylanase, basal diet + 100 g/t of xylanase.

**Table 7 animals-13-00549-t007:** The effects of dietary NSPases and phytase on the microbial alpha diversity indices of growing–finishing pigs.

Items	Control	Phytase	β-Mannanase	β-Glucanase	Xylanase	*p*-Values
OTUs	604.3 ± 10.23	517.8 ± 35.52	567.2 ± 28.51	509.8 ± 18.41	529.0 ± 39.38	0.139
Chao	719.3 ± 5.56	644.2 ± 36.67	671.4 ± 28.8	635.4 ± 23.03	625.7 ± 43.01	0.219
Shannon	4.60 ± 0.10 ^a^	3.58 ± 0.23 ^c^	4.22 ± 0.09 ^ab^	3.69 ± 0.14 ^bc^	4.03 ± 0.18 ^bc^	0.004
Simpson	0.03 ± 0.00 ^c^	0.11 ± 0.03 ^a^	0.04 ± 0.00 ^bc^	0.07 ± 0.01 ^ab^	0.05 ± 0.01 ^ab^	0.001
ACE	700.3 ± 6.31	642.6 ± 35.72	671.8 ± 30.27	623.7 ± 17	618.2 ± 43.1	0.276

Data are expressed as means ± SEM (n = 6). Means within a row with different superscripts are significantly different (*p* < 0.05). Control, basal diet; phytase, basal diet + 100 g/t of phytase; β-mannanase, basal diet + 40 g/t of β-mannanase; β-glucanase, basal diet + 100 g/t of β-glucanase; xylanase, basal diet + 100 g/t of xylanase.

## Data Availability

All data used in the current study are available from the corresponding author on reasonable request.

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
