# Peer review of "The Effects of Phytase and Non-Starch Polysaccharide-Hydrolyzing Enzymes on Trace Element Deposition, Intestinal Morphology, and Cecal Microbiota of Growing–Finishing Pigs"

_animals, 2023, doi:10.3390/ani13040549_

Round 1

Reviewer 1 Report (Previous Reviewer 1)

I do not have any further comments.

Reviewer 2 Report (Previous Reviewer 2)

Dear Authors thank you for improving manuscript. In my opinion it is interesting and could be published in the current form. 

This manuscript is a resubmission of an earlier submission. The following is a list of the peer review reports and author responses from that submission.

Round 1

Reviewer 1 Report

This study explored the effects of dietary phytase and 3 different NSPases supplementation on trace minerals deposition in tissues, genes expression related to trace minerals absorption, small intestinal morphology, and cecal microbiota compositions in growing-finishing pigs. The animal trial was well designed and performed, plenty of data were collected, and the results were well-illustrated and discussed. The findings in the current study will be beneficial to practical pig production. I only have a few minor comments.

1. L77-79, more information is need for the 3 NSPases, e.g., sources, enzyme activity.

2. L83-84, six pigs per treatment group were randomly selected and sacrificed?

3. Table 1, CP content in soybean meal and form of the crystalline AAs (L-Lys-HCl, and purity?) were needed.

4. Table 1, Try should be Trp.

5. Part 2.7, analysis methods of microbial sequencing data should also be briefly described here.

6. Figure 4, the multiple comparison method in this figure was different from those dealing with other indexes, and those should be stated and clarified.

Author Response

Dear reviewers,

Thank you very much for your letter and the comments on our manuscript. Now we have revised the manuscript according to your comments. All revisions were marked red in the manuscript.

Response to Reviewer 1 Comments

  1. L77-79, more information is need for the 3 NSPases, e.g., sources, enzyme activity.

Response: “The exogenous enzymes (phytase, β-mannanase, β-glucanase, and xylanase) were pro-vided by the Wuhan Sunhy Biology Co., Lid. (Wuhan, China). The phytase, β-mannanase, β-glucanase, and xylanase contained 5000U of phytase/g, 1,000 U of β-mannanase/g, 5,000 U of β-glucanase/g, and 10,000 U of xylanase/g, respectively.” has been added in Part 2.1.

  1. L83-84, six pigs per treatment group were randomly selected and sacrificed?

Response: Yes, that is correct, each group has ten pigs. At the end of the experiment, six pigs with the average body weight per group were randomly selected and euthanized.

  1. Table 1, CP content in soybean meal and form of the crystalline AAs (L-Lys-HCl, and purity?) were needed.

Response: “CP content in soybean meal and form of the crystalline AAs (L-Lys-HCl, and purity?)” were added in Table.1.

  1. Table 1, Try should be Trp.

Response: Table 1 “Try” was revised as “Trp”.

  1. Part 2.7, analysis methods of microbial sequencing data should also be briefly described here.

Response: “Raw fastq files were demultiplexed, quality-filtered using QIIME (version 1.17). Operational Units (OTUs) were clustered with 97% similarity cutoff using UPARSE (version 7.1 http://drive5.com/uparse/) and chimeric sequences were identified and removed using UCHIME. The taxonomy of each 16S rRNA gene sequence was analyzed by RDP Classifier (http://rdp.cme.msu.edu/) against the silva (SSU115)16S rRNA database using confidence threshold of 70%. To compare differences between dietary treatments, data for α-diversity, bacterial phyla and genera were subjected to ANOVA.” was added in Part 2.6.

  1. Figure 4, the multiple comparison method in this figure was different from those dealing with other indexes, and those should be stated and clarified.

Response: The multiple comparison in Figure 4 was conducted with ANOVA. “To compare differences between dietary treatments, data for α-diversity, bacterial phyla and genera were subjected to ANOVA.” were added in Part 2.6.

Reviewer 2 Report

Review:  The effects of phytase and non-starch polysaccharide hydrolyzing enzymes on trace element deposition, intestinal morphology and caecal microflora of growing-finishing pigs.

The aim of study was to evaluate the effects of the supplementations of NSPases (β-mannanase, β-glucanase, and xylanase) and phytase to corn-soybean meal-based diets on growth performance, trace element deposition, intestinal morphology, and caecal microflora of growing-finishing pigs. The subject is interesting but the literature with using different enzymes is well known. The authors did not highlight the innovations of the research – it must by added in the manuscript.

Abstract

L. 34 Eubacterium ruminantium – italic?

Introduction

L.50, 55, 98, 120, 281, 303 – lack of space

L.53 – mannans

L.57 – phytate

L. 59 – it is not suggested. It is well known!

L.61 – partially hydrolyze

M&M

Why did you use this kind of dosing? Were that commercial enzymes? What was producer`s recommendations??

Why pigs in this age were chosen?

In what form pigs were offered diets?

Table 1 – amino acids description – D? L? 97% or others?  Ingredients – the lack of unit.

L. 121 – what is H&E?

Zn, Fe and Cu were also present in the premix and they were in the non-organic form. Did you calculate the total amount of these minerals in the total diet? Could enzyme change the environment of DT and effect better utility of these minerals from their salts?

Results

L. 144, 159 -  in table title – remove the point.

L 149 – why n=6?  In my opinion it should be 10.

L. 173-4 – information (ileum, duodenum, jejunum) appeared twice in the graph and in graph description.

L. 189 – n=6

Table 5, 6 – in some line the values are hard to read – maybe you should reduce some decimal points.

When you describe groups sometimes you use low (in text) and sometime big letters (under tables)– please use one description.

L. 243-7 – different font size

Figure 5. n--=6??

L. 257-60 – it is not clear. They also had no positive affect.

Discussion

L. 258-290 – genes expression and minerals content should be better described in one part of discussion.

L. 300 - Luo, J et al. – should be – Luo et al. [29]…

L. 302 – Moita….- improve as above.

L. 306 – what is the relation in the amounts of enzymes used in your experiments in g/t with data as xylanase (5000 FTU/t) and phytase (10,000 BXU/t)? Is this similar level??

Conclusion should be more specific.

References 7, 8, 9, 13, 15, 16, 17, 18, 22, 25, 27, 31, 33, 35, 37, 38 must be checked and improved.

Round 2

Reviewer 1 Report

1, Either CP 43% or CP 46% SBM was used in feed production in practice. I did not see any lablels for CP 42.5% SBM.

2, The microbiota sequencing data were usually not followed the normal distribution, thus nonparametric test was usually used. The authors stated that ANOVA was used for such data - please prove these data were normally distributed - otherwise the results may be incorrect.

Author Response

Dear reviewers,

Thank you very much for your letter and the comments on our manuscript. Now we have revised the manuscript according to your comments. All revisions were marked red in the manuscript.

Response to Reviewer 1 Comments

1, Either CP 43% or CP 46% SBM was used in feed production in practice. I did not see any lablels for CP 42.5% SBM.

Response: It's my fault. “42.50%” was revised as “43%”.

2, The microbiota sequencing data were usually not followed the normal distribution, thus nonparametric test was usually used. The authors stated that ANOVA was used for such data - please prove these data were normally distributed - otherwise the results may be incorrect.

Response: This is my writing error. The the 16 rRNA sequencing data were analyzed by a nonparametric test. Part 2.7 “Data are presented as mean value ± standard error of the mean. Statistical analyses and graphs were performed using one-way ANOVA analysis followed by Tukey’s honestly significant difference test by SPSS (version 26) [23,24]. Differences were con-sidered significant when p < 0.05 and noted with different superscript letters.” was revised as “All statistical analysis were performed using IBM SPSS 22.0 software (SPSS Inc, Chicago, IL, USA) except for microbiome analysis. The differences among experimental treatments were performed using one-way ANOVA analysis followed by Tukey’s honestly significant difference test [23,24]. After nonparametric tests, the the 16 rRNA sequencing data were analyzed by a Kruskal-Wallis analysis to determine significant differences. Data are presented as mean value ± standard error of the mean (SEM) unless otherwise noted. Differences were considered significant when p < 0.05 and noted with different superscript letters.”